# MGMT-Methylation in Non-Neoplastic Diseases of the Central Nervous System

**DOI:** 10.3390/ijms22083845

**Published:** 2021-04-08

**Authors:** Sarah Teuber-Hanselmann, Karl Worm, Nicole Macha, Andreas Junker

**Affiliations:** 1Institute of Neuropathology, University Hospital Essen, D-45147 Essen, Germany; Sarah.Teuber@uk-essen.de (S.T.-H.); Nicole.Macha@uk-essen.de (N.M.); 2Institute of Pathology, University Hospital Essen, D-45147 Essen, Germany; Karl.Worm@uk-essen.de

**Keywords:** DNA-methylation, MGMT promoter methylation, multiple sclerosis, progressive multifocal leucencephalopathy (PML), central pontine and exptrapontine myelinolysis, Wallerian degeneration

## Abstract

Quantifying O^6^-methylguanine-DNA methyltransferase (MGMT) promoter methylation plays an essential role in assessing the potential efficacy of alkylating agents in the chemotherapy of malignant gliomas. MGMT promoter methylation is considered to be a characteristic of subgroups of certain malignancies but has also been described in various peripheral inflammatory diseases. However, MGMT promoter methylation levels have not yet been investigated in non-neoplastic brain diseases. This study demonstrates for the first time that one can indeed detect slightly enhanced MGMT promoter methylation in individual cases of inflammatory demyelinating CNS diseases such as multiple sclerosis and progressive multifocal leucencephalopathy (PML), as well as in other demyelinating diseases such as central pontine and exptrapontine myelinolysis, and diseases with myelin damage such as Wallerian degeneration. In this context, we identified a reduction in the expression of the demethylase TET1 as a possible cause for the enhanced MGMT promoter methylation. Hence, we show for the first time that MGMT hypermethylation occurs in chronic diseases that are not strictly associated to distinct pathogens, oncogenic viruses or neoplasms but that lead to damage of the myelin sheath in various ways. While this gives new insights into epigenetic and pathophysiological processes involved in de- and remyelination, which might offer new therapeutic opportunities for demyelinating diseases in the future, it also reduces the specificity of MGMT hypermethylation as a tumor biomarker.

## 1. Introduction

O^6^-methylguanine-DNA methyltransferase (MGMT) is an important constitutively active enzyme which is expressed in every human cell, playing a pivotal role in the cellular defense against the toxicity of alkylating substances by removing methyl groups, particularly O^6^-methylguanine residues, thereby repairing alkylated DNA and preventing mismatch errors during DNA replication [1]. Hypermethylation of the MGMT promoter region results in gene silencing, accompanied by decreased DNA repair, an effect that is seen in various tumors, including lung carcinoma, head and neck carcinomas, lymphoma, colorectal carcinoma, melanoma [1,2,3,4,5], as well as glioma [6] and particularly oligodendrogliomas [7]. MGMT hypermethylated gliomas are much more responsive to therapies with alkylating chemotherapeutics such as temozolomide (TMZ) than those without MGMT promoter hypermethylation [6,8]. MGMT promoter methylation seems to be at least partly the result of isocitrate dehydrogenase (IDH) mutations in gliomagenesis, since such gain-of-function mutations lead to the increased production of 2-hydroxyglutarate instead of alpha-ketoglutarate, which itself is an important co-factor for the proteins of the ten-eleven-translocation (TET) methylcytosine dioxygenases family such as TET1 and TET2 [9,10]. The TET family proteins are important DNA demethylases, and reduced levels of their co-factor alpha-ketoglutarate result in reduced enzyme activity, followed by globally increased DNA methylation in these cells, a phenomenon named G-CIMP [11].

In brain tissues with no evidence of any pathological alterations, the methylation rates of cytosine residues of the MGMT promoter region are described as not exceeding 3–4%, which in turn does not affect MGMT protein expression (so-called non-methylated MGMT promoter) [1,12]. Outside the central nervous system (CNS), MGMT promoter hypermethylation has been described in inflammatory diseases, especially in chronic inflammatory diseases of the gut, the liver or the colon [13,14]. In such cases, infections with oncogenic viruses, such as hepatitis-C-virus (HCV), Epstein–Barr virus (EBV) and hepatitis B virus (HBV) seem to be the critical event for MGMT promoter methylation [15,16,17,18]. Despite this knowledge, it has not yet been investigated whether MGMT promoter hypermethylation is present in non-neoplastic diseases of the CNS. The aim of our study is to investigate whether MGMT promoter methylation is a phenomenon that is restricted to neoplasms in the CNS, or whether it could be detected in other non-neoplastic CNS pathologies as well. Our study is, therefore, the first to analyze MGMT promoter methylation in a variety of non-neoplastic CNS diseases, and we report cases of variable MGMT hypermethylation in infectious, inflammatory and demyelinating CNS diseases, as well as in those resulting in damage to the myelin sheath.

## 2. Results

### 2.1. The MGMT Promoter Is Variably Methylated in Various Non-Neoplastic CNS Diseases

We investigated autopsy and biopsy samples of healthy and diseased brains in terms of MGMT promoter methylation. Our analysis included brains without evidence of any pathological changes (“healthy controls”), as well as brains with infectious non-demyelinating diseases, i.e., bacterial, mycotic, inflammatory, viral (induced by HSV or HIV), or parasitic (toxoplasmosis), those with inflammatory-demyelinating diseases, i.e., multiple sclerosis (MS) and progressive multifocal leukoencephalopathy (PML), as well as those with non-neoplastic and non-inflammatory CNS conditions that damage the myelin sheaths, i.e., central pontine myelinolysis (CPM), extrapontine myelinolysis (EPM) and Wallerian degeneration. The degree of MGMT promoter methylation was compared with that in healthy controls (Table 1, Figure 1) as well as in hypermethylated gliomas (Appendix A). The mean percentage of methylated cytosine residues, measured at five CpG sites in each sample, was 5.6% in healthy controls (SD 0.88). The highest methylation rate in our controls was 6.6%. A sample was defined as “hypermethylated” if the mean methylation rate of all five CpG sites exceeded 7.7%, which was equivalent to the third quartile of the methylation rate in all non-neoplastic samples (Table 1, Figure 1).

The methylation rate of the MGMT promotor varied significantly across all samples, including those that were non-methylated as well as those with prominent hypermethylation. We identified MGMT promoter hypermethylation not only in samples from patients with MS and PML, but also in those with metabolic and degenerative diseases, such as EPM, CPM and Wallerian degeneration (Table 1, Figure 1). Isolated samples did exhibit methylation rates similar to those in moderately to highly hypermethylated gliomas (Appendix A). The hypermethylated samples could not be assigned to distinct pathological conditions, and, in addition, diseases associated with hypermethylation also included non-methylated samples.

Interestingly, no samples from the group of infectious non-demyelinating diseases had MGMT promoter methylation rates higher than those in healthy controls, which stands in contrast to the findings in extra-CNS samples. In fact, all samples with hypermethylation were found in the groups with inflammatory demyelinating diseases or non-inflammatory metabolic and degenerative diseases accompanied by damaged myelin sheaths (Figure 1).

The extent of hypermethylation showed no significant correlations with the age or sex of the patients (age: Pearsons r = −0.03; *p* = 0.82; sex: Pearsons r = 0.15; *p* = 0.18) or with the total number of apoptotic cells measured by immunohistochemical staining for caspase-3 (Pearsons r = 0.23, *p* = 0.24). These results are shown in Appendix A. In order to rule out any influence of cause of death on the extent of methylation, we categorized our autopsy samples into five groups concerning their cause of death, namely (1) multiorgan failure (*n* = 7), (2) sepsis (*n* = 4), (3) cardiac reasons (*n* = 5), (4) pulmonary reasons (*n* = 14) and (5) others (*n* = 13). The means of methylation of those groups were compared to each other via ANOVA afterwards. We did not find any significant differences between the groups (*p* > 0.05). Therefore, the (reason of) death itself does not seem to affect MGMT methylation (Appendix A).

It is worth mentioning that MGMT promoter methylation rates differed significantly between samples obtained from MS patients by biopsy or autopsy (biopsy: mean 4.13%, SD 0.8 vs. autopsy: mean 8.72%, SD 9.8; *p* < 0.01). All deceased MS patients whose autopsy samples we analyzed had suffered from long-standing chronic MS, whereas most patients who had been diagnosed with MS by brain biopsy had a short history of symptoms and disease course (i.e., a few months maximum; the only exception was MS sample 8). In each case, the biopsy had been performed because a tumor had been suspected. Therefore, it seems plausible that MGMT promoter methylation is associated with long-standing and chronic MS disease rather than with more acute and active forms. Of note, the autopsy samples were from MS patients significantly older than those who had had a brain biopsy (autopsy: mean 56.1 years, SD 11.5 vs. biopsy: mean 35.6 years, SD 15.4; *p* < 0.01). However, since there was no association between age and MGMT promoter methylation across all analyzed samples (Appendix A) it is much more likely that the activity status of the disease is responsible for the differences in MGMT methylation.

Since 50% of the hypermethylated samples (ten out of 20 samples) were linked to non-inflammatory diseases, such as CPM, EPM and Wallerian degeneration, an association of MGMT promoter methylation to inflammatory infiltrate per se and, in particular, to specific inflammatory cells (e.g., lymphocytes, granulocytes, plasma cells) was not expected.

However, since injured axonal networks display the fundamental commonality of all hypermethylated samples, we suspected a link between axonal damage and MGMT promoter methylation. SMI31 is a marker for phosphorylated neurofilaments which display the integrity of axonal networks [19]. We therefore stained all samples against SMI31 and correlated the density of SMI31-positive phosphorylated neurofilaments to MGMT promoter methylation rate. Nonetheless, we were unable to verify our hypothesis in this investigation (Pearsons r = 0.05; *p* = 0.78). However, this negative result might be due to relatively small differences in methylation levels compared with healthy controls, as well as a heterogeneity of hypermethylation and non-methylation within one disease entity.

Since the degree of hypermethylation was relatively low, we next wanted to investigate whether the slight differences might be reflected in lowered MGMT mRNA and protein expression levels. To begin with, we attempted to quantify MGMT mRNA levels in formalin fixed paraffin embedded (FFPE) samples from healthy controls and those with hypermethylation using qPCR. Suitable results were only obtained for a few samples (healthy control (control 1–4); CPM/EPM: CPM 2, Wallerian degeneration: WAL 2, WAL 3), and a reliable statistical assessment was not possible due to the small sample size. Nonetheless, MGMT mRNA levels tended to be lower in hypermethylated samples compared with healthy controls (Wallerian degeneration: 1.36-fold lower levels, CPM/EPM: four-fold lower levels) (Appendix A).

In a next step, all samples were stained immunohistochemically against MGMT for further analysis of MGMT protein expression levels. For the quantification of MGMT expression in glial cells, we automatically measured the numbers of MGMT-positive glial cells per mm^2^. Of note, small differences in methylation rates of the MGMT promoter resulted in significantly lower MGMT protein expression levels (Figure 2). Healthy controls (Figure 2C) showed the constitutive expression of MGMT in almost all glial cells, i.e., astrocytes and particularly in oligodendrocytes, as well as in neurons, while the staining intensity of glial cells was significantly reduced in all samples with hypermethylated MGMT promoter (Figure 2D–H) (*p* < 0.01, Wallerian degeneration not significant due to small sample size). Indeed, the hypermethylated samples showed only scattered MGMT-expressing glial cells in the areas of the lesions (Figure 2D–H), while the neurons showed stable staining intensities across all diagnostic groups (Figure 2C–H, insets; for quantification see Appendix A), serving as an internal positive control and thereby ruling out staining artifacts. The number of MGMT-positive glial cells was found to be independent of the extent of myelination in MS lesions (chronic inactive plaque (CIAP) vs. remyelinated shadow plaques) (Figure 2D,E) (no significant difference between the groups).

### 2.2. The Protein Expression of Demethylase TET1 Is Associated with MGMT Promoter Methylation

After realizing that MGMT promoter hypermethylation is a frequent phenomenon in non-neoplastic CNS diseases, the question arose regarding the underlying mechanism. Enzymes of the TET protein family are fundamental players in stabilizing DNA methylation patterns by regulating DNA demethylation. TET1 and TET2 are important members of this enzyme family, which are expressed in brain tissue [20]. Bearing in mind that DNA-hypermethylation is often a result of an imbalance of DNA methylation and demethylation which can be based on reduced TET1 protein activity as shown in gliomas, we investigated the protein levels of TET1 and TET2 via immunohistochemistry. Since MGMT expression differences have only been demonstrated in glial cells but not in neurons (see Appendix A) and since diseases with disintegration of the myelin sheaths are located in the white but not the grey matter, we analyzed and in particular quantified TET expression in glial cells only.

Healthy controls showed a broad spectrum of nuclear TET1-staining intensity with numerous highly positive glial cells (i.e., high TET1-expressing cells) as well as few faintly positive glial cells (i.e., low TET1-expressing cells) (Figure 3A,C). Hypermethylated samples from various conditions, however, showed numerous glial cells with only weak or even completely absent TET1 positivity and only small numbers, if any, of highly positive glial cells (Figure 3A, D–H). Since neurons showed stable staining intensities across all diagnostic groups (Figure 3D–H, insets), they served as an internal positive control and hence staining artifacts were ruled out. Therefore, we were able to demonstrate an overall reduction in TET1 protein expression levels in the different kinds of conditions showing MGMT promoter hypermethylation (Figure 3A–H). Notably, TET1 expression levels were independent of the degree of myelination in MS lesions (CIAP vs. shadow plaque) (Figure 3D,E), similar to the staining results of MGMT (Figure 2).

It is noteworthy that the percentage of high TET1-expressing glial cells correlated negatively with the MGMT promoter methylation rate (Figure 3B, lower panel), while conversely, that of low TET1-expressing glial cells (Figure 3B, upper panel) was positively correlated with it.

In contrast to TET1, there were no significant correlations between TET2 expression and MGMT promoter methylation (Appendix A).

### 2.3. Calpain-1 Does Not Regulate TET1 Expression in Hypermethylated Non-Neoplastic CNS Diseases

Since we hypothesized that a reduced TET1 expression is at least partly the basis of MGMT promoter methylation in non-neoplastic CNS diseases, we again questioned the underlying mechanism. A previous study has shown that TET1 and TET2 expression and activity can be regulated by Calpain-1 via Calpain-1-mediated cleavage [21]. Furthermore, associations of Calpain-1 with axonal damage and myelin degradation have been described several times in the literature [22,23,24,25]. Against this background, we investigated Calpain-1 expression in healthy control samples and hypermethylated samples from any condition by immunohistochemistry (Figure 4).

Notably, we did not find any difference in Calpain-1 expression in neurons as well as in glial cells between hypermethylated and healthy control samples (Figure 4A–H) (*p* > 0.05). Additionally, there were thus no correlations between Calpain-1 and TET1 (Pearsons r = 0.28; *p* = 0.26) or between Calpain-1 and MGMT promoter methylation (Pearsons r = −0.3, *p*= 0.16).

## 3. Discussion

MGMT is a protein that plays an important role in DNA repair and cellular defense against toxic agents such as alkylating substances. It catalyzes the transfer of methyl groups from mainly O^6^-methylguanine but also O^4^-methylguanine to cysteine residues of its own molecule, thereby repairing damaged DNA [1]. MGMT promoter methylation is a common phenomenon in inflammatory diseases outside the CNS and is often associated with oncogenic viruses (e.g., HBV, EBV, HCV) [15,16,17,18]. Moreover, MGMT promoter hypermethylation had been described in precancerous lesions such as colitis ulcerosa and Crohn’s disease but also in gastritis [13,26] as well as in various cancers, including colorectal carcinoma, lung carcinoma, lymphoma, melanoma, and, in the CNS in glioma, particularly oligodendroglioma but also in IDH-mutated as well as in IDH-wildtype astrocytoma [5,6]. Previously, it was postulated that DNA hypermethylation is a phenomenon attributed to cancers or to malignant transformation of precancerous lesions and that consequently it could be used as a tumor biomarker [27]. Up to now, only a few studies have focused on MGMT promoter methylation in the CNS beyond cancerous brain tissues. Hsu et al. investigated healthy brains and brain biopsies obtained during epilepsy surgery and found that none of their samples exhibited MGMT promoter hypermethylation [12]. A single study analyzed the promoter methylation rates of different DNA repairing enzymes, including MGMT, in Alzheimer’s disease and also did not find the MGMT promoter to be hypermethylated [28]. Thus, both studies underscored the hypothesis [27] that MGMT methylation can be attributed to CNS neoplasms rather than to non-neoplastic CNS diseases. Nonetheless, other CNS diseases, particularly of infectious and inflammatory nature have not yet been studied. To our knowledge, we were the first to show that MGMT promoter hypermethylation takes place in a variety of non-neoplastic CNS diseases. While we were not able to attribute MGMT methylation to single disease entities or distinct pathogens, it is noteworthy that we did not find any associations with infectious non-demyelinating diseases (i.e., bacterial or mycotic abscesses, toxoplasmosis, viral encephalitis), a fact that is in contrast to the findings outside the CNS. However, we found MGMT promoter hypermethylation in the samples from patients with diseases associated with disintegrated myelin sheaths, i.e., inflammatory and demyelinating (MS and PML), as well as non-inflammatory metabolic or degenerative CNS diseases (CPM/EPM/Wallerian degeneration). These results, of course, reduce the hypothesized specificity of MGMT promoter methylation for neoplastic processes [27].

Our results show that the spectrum of MGMT promoter methylation in healthy controls is relatively narrow with a range of 4 to 6.6 percent, which is in accordance with those of Christmann et al. and Hsu and colleagues [1,12]. They underscore the finding that healthy brain tissue is apparently never hypermethylated and that MGMT promoter methylation is only associated with severe brain diseases. The fact that MGMT hypermethylation was especially found in autopsy samples from patients with long-standing, chronic MS, rather than in biopsies where the disease was more active in nature, leads to the hypothesis that MGMT promoter methylation might be associated with later disease stages accompanied by axonal damage. This hypothesis is underscored by various studies that linked MGMT methylation to later disease stages. For example, Alvarez et al. demonstrated MGMT hypermethylation in chronic, but not in early stages of *Helicobacter pylori*-associated gastritis [29], and the study of N. Zekri et al. highlighted that long-standing chronic Hepatits C infection with progression to hepatocellular carcinoma is accompanied by promoter methylation of APC gene in early, and that of MGMT in later disease stages [30]. Furthermore, MGMT methylation is seen in COPD and lung cancers due to chronic tobacco abuse [31].

This hypothesis of MGMT methylation in later disease stages could explain the great heterogeneity in terms of MGMT methylation levels in the individual samples from our cohort. This mixture of hypermethylation and non-methylation within one disease entity (as observed in our samples) is a well-known phenomenon, which has been described in chronic hepatitis C infected and fibrotic livers [15], Hodgkin lymphomas [17], brain metastases of various cancers (reviewed in [1]), glioblastoma [32] as well as in our glioma samples (Appendix A). This underscores that MGMT promoter methylation shows a wide range of methylation within one disease entity, while the exact cause of this broad spectrum remains unknown. Nonetheless, we are aware that our total numbers of samples from the various disease entities were small and that our results might thus not be generalizable. As a consequence of our relatively small sample sizes, MGMT promoter methylation rates did—of course—not reach statistical significance, since this would only be possible if we could have investigated hundreds of samples. Bearing in mind that those disease entities investigated here are relatively rare and biopsy or autopsy material is even rarer, we really believe that—although not reaching statistical significance—our collection is a relatively big one and that our results are anyhow of great interest.

The differences in methylation between the hypermethylated samples and those of healthy controls were often relatively small. This might have been due to the fact that we used whole slides for our analysis and did not perform any macrodissection of the lesions or microdissection of individual cells. While we believed that analyzing the lesion as well as the surrounding brain microenvironment would give more information than assessing the lesion or individual cells alone, this approach may have diminished the magnitude of our detected effects. Nonetheless, we were able to show reduced MGMT mRNA and especially protein expression levels in hypermethylated samples despite the low-level methylation differences. This is a well-known effect since small differences in DNA methylation are accompanied by altered protein expression levels, as has been shown in rheumatoid arthritis and in the frontal cortices of Alzheimer’s disease patients [33,34].

Chronic gastritis induced by *Helicobacter pylori* showed MGMT promoter hypermethylation based on accelerated NF-κB signaling [35]. The pro-inflammatory NF-κB pathway plays a pivotal role in various inflammatory diseases [36]. In MS, NF-κB is essentially involved not only in the activation of peripheral immune cells but also in reactive processes of brain-derived cells [37]. Upregulated NF-κB activity has also been detected in PML and Wallerian degeneration [38,39].

A fundamental similarity of all hypermethylated samples is the damage to the myelin sheath. Oligodendrocytes are glial cells that are responsible for producing myelin and for effective remyelination. Different DNA methylation patterns are involved in oligodendrocyte differentiation, myelin production and remyelination after injury. High TET1 expression levels seem to be required for efficient remyelination after myelin damage [20]. A recent study reported that demyelinated lesions in the hippocampus of MS patients showed upregulated levels of DNA methyl transferases (DNMTs)—enzymes that are responsible for DNA methylation—while TET levels were downregulated [40]. We have also found reduced TET1 protein expression levels in all our samples with MGMT promoter hypermethylation, leading to the hypothesis that there is an imbalance between DNA methylation and DNA demethylation at least by reduced DNA demethylation. Whether DNA methylation is additionally enhanced by upregulated DNMT expression levels as already shown for MS [40] has to be investigated in future studies. Against this background, it is possible that the MGMT promoter, as well as general DNA methylation in other DNA regions, displays an essential epigenetic mechanism for the defective myelin regeneration in CNS diseases with myelin damage and loss. Indeed, it was shown that promoters of genes such as BCL2L2 and NDRG1, both genes that regulate oligodendrocyte survival, are hypermethylated in normal appearing white matter of MS patients, and that consequently the proteins are less expressed, which in turn leads to accelerated oligodendrocyte apoptosis, accompanied by less efficient remyelination [41].

Further and still larger scale studies with whole genome methylation analyses should be performed to better understand the influence of DNA methylation on de- and remyelination processes in MS and other demyelinating diseases. It is known that cell type-specific methylation patterns exist [20]. It would, therefore, be important to also investigate the relation of oligodendrocytes or astrocytes to possible DNA hypermethylation in further studies.

In summary, we have shown for the first time that MGMT hypermethylation is indeed found in non-neoplastic CNS diseases showing damage of the myelin sheath due to various conditions. Thus, we demonstrated that MGMT methylation is not restricted to neoplasms or strictly associated to distinct pathogens, as well as oncogenic viruses or bacteria. Therefore, the capability of MGMT hypermethylation to serve as a biomarker for neoplasms or as indicator of malignant transformation of precancerous lesions is highly reduced by our results. The fact that MGMT methylation is found in chronic processes that lead to myelin loss and consequently to axonal damage, shed light into possible epigenetic and pathophysiological processes involved in demyelination and might thus offer new therapeutic opportunities in the future.

## 4. Materials and Methods

### 4.1. Human Brain Tissue

The investigations were performed on formalin-fixed and paraffin-embedded (FFPE) autopsy or biopsy tissue samples from patients with PML (*n* = 10), MS (*n* = 28), toxoplasmosis (*n* = 6), cytomegalovirus infection (*n* = 1), HSV1 encephalitis (*n* = 1), HIV infection (*n* = 2), with mycosis and encephalitis (*n* = 4), brain abscess (*n* = 3), central pontine/extrapontine myelinolysis (*n* = 8), Wallerian degeneration (*n* = 3), or glioma (*n* = 71) and *n* = 8 samples from healthy controls with no pathological changes. See Table 1 and Appendix A for details on type of sample (biopsy/autopsy) and demographics (age, sex, cause of death, disease duration). The biopsy samples of the above-mentioned disease entities were obtained for diagnostic purposes. All of the autopsied MS patients had suffered from long-term chronic MS whereas the MS biopsies (with the exception of MS biopsy #8) had been performed in patients with a disease course of only a few months to exclude a tumor.

*n* = 18 of the MS autopsy samples were obtained from the Netherlands Brain Bank (NBB), where they had been evaluated and the diagnoses were confirmed by one author (A.J.). All other tissue samples were obtained from the archive of the Institute of Neuropathology of the University Hospital Essen; they represent all cases of PML, MS, toxoplasmosis, CMV, HSV1, HIV, brain mycosis, CMP/EPM and Wallerian degeneration that had been diagnosed over the last three decades in our institute and that owned enough biopsy/autopsy material to be fully analyzed in our study. The clinical histories of the patients were evaluated for the study as far as possible.

### 4.2. Histology and Immunohistochemistry

All investigations were performed on whole slide sections of 1 µm thickness. In addition to standard staining with hematoxylin––eosin (HE) (not shown), immunohistochemical staining was performed with antibodies against MGMT, TET1, TET2, Calpain-1, SMI31 and Caspase-3 according to standard procedures. Pretreatments and antibody dilutions were carried out as described in Table 2.

In brief, the endogenous peroxidase activity was blocked by incubating the sections in 3% H_2_O_2_ in PBS, followed by a blocking step with 10% fetal calf serum in PBS for ten minutes at room temperature and by incubation with the primary antibody for one hour at room temperature. The sections were then incubated with the secondary antibody (biotin-labelled antibody). Finally, the immunohistochemical stain was developed with 3,3′-diaminobenzidine (DAB). Cell nucleus counterstaining was performed with hematoxylin. Some sections were stained using the DAKO Autostainer Plus. In these cases, the ZytoChemPlus HRP Polymer System (Mouse/Rabbit) (REF: POLHRP-100) was used for detection.

The stained sections were first digitized using a Leica slide scanner. From the scanned files, five areas with an edge length of 500 µm were extracted from all regions of interest and analyzed using Image J [42]. After adjusting hue, saturation and brightness, the “color threshold” was adjusted so that colored particles or colored areas could be determined using the “Analyze Particles” function [43]. To analyze the different saturation levels of TET1-stained images, each image was automatically analyzed 256 times with different saturation thresholds. The total maximum saturation found over all images was set equal to 100%.

### 4.3. cDNA Synthesis and Quantitative PCR

Total RNA was extracted from 15 FFPE tissue sections (5 µm thickness) with the QIAamp DNA FFPE Tissue Kit (Qiagen, Hilden, Germany). The transcription of RNA into complementary DNA (cDNA) was performed with the high-capacity cDNA reverse transcription kit (Thermo Fischer Scientific, Dreieich, Germany) according to the manufacturer’s instructions. 200 ng RNA aliquots were used for each reaction (20 µl). The qPCR reaction was performed with the qPCR core kit (from Eurogenetec, Cologne, Germany). GAPDH was used as the housekeeping gene [44]. Transcripts were analyzed with TaqMan assays against MGMT (Assay number: Hs01037698_m1, Thermo Fisher Scientific, former Applied Biosystems, Dreieich, Germany).

### 4.4. MGMT Promoter Methylation Analysis

MGMT promoter methylation analysis was performed via pyrosequencing. This method is based on the bisulfite-dependent conversion of cytosine residues to uracil, which does not convert 5-methylcytosine residues. After amplification and bisulfite conversion of the template DNA and pyrosequencing, the percentages of non-bisulfite converted (i.e., methylated) cytosine residues can be analyzed at different CpG sites of the MGMT promoter.

DNA was extracted from whole tissue slices using the Maxwell RSC DNA FFPE Kit (AS1720, Promega, Mannheim, Germany) according to the manufacturer’s instructions. The bisulfite conversion was then carried out using the EpiTech Plus DNA Bisulfite Kit (Qiagen, Hilden, Germany) following the manufacturer’s instructions. For the quantification of CpG methylation in regions +17 to +39 in the MGMT gene, 500 ng of bisulfite-converted DNA was treated according to the manufacturer’s instructions for the PyroMark Q24 CpG MGMT kit (Qiagen, Hilden, Germany). The locations of the five analyzed CpG sites were as follows: Chr10:131265507, 131265514, 131265519, 131265522 and 131265526 (Genome Biology 17:55). Analysis of not more than five CpG sites of MGMT promoter in glioma is a common practice in routine neuropathological diagnostics (for review see [45]). In order to reach better comparability between MGMT methylation rates of non-neoplastic and neoplastic CNS diseases, we investigated the same five CpG sites that are routinely tested in gliomas in our institute.

### 4.5. Statistical Analysis

GraphPad Prism 5.0 was used for statistical analysis and evaluation. The Mann–Whitney U-test was used to compare independent groups. The correlation between groups was calculated as Pearson’s r. A *p*-value of <0.05 was considered statistically significant and <0.01 as highly significant.

## 5. Conclusions

To conclude, MGMT methylation occurs in a variety of CNS pathologies and has to be interpreted carefully in the context of clinical and histological conditions.

## Figures and Tables

**Figure 1 ijms-22-03845-f001:**
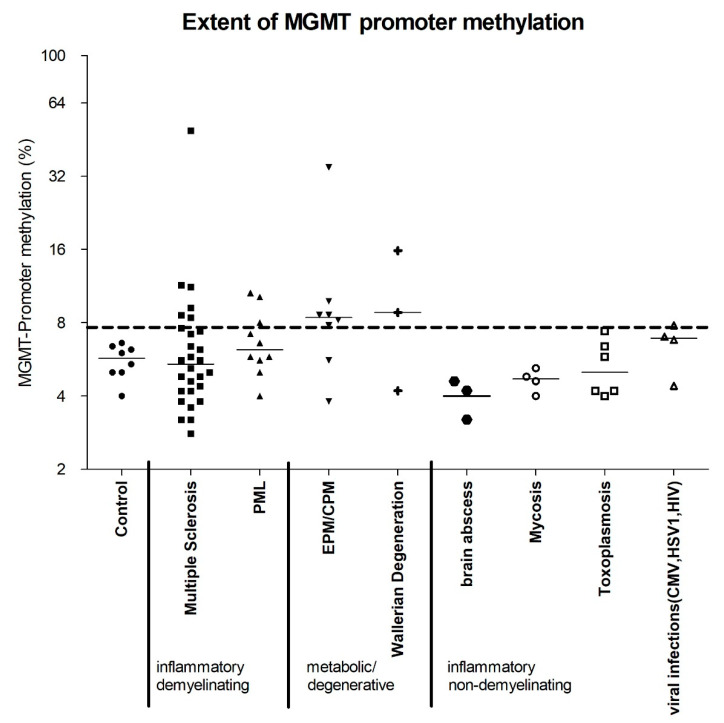
Summarizing evaluation of MGMT promoter methylation of inflammatory and metabolic/degenerative CNS diseases. MGMT promoter methylation status was assessed at five different CpG sites in the MGMT promoter for each demonstrated sample (control *n* = 8, MS *n* = 28, PML *n* = 10, EPM/CPM *n* = 8, Wallerian degeneration *n* = 3, brain abscess *n* = 3, mycotic encephalitis *n* = 4, toxoplasmosis *n* = 6, CMV *n* = 1, HSV1 *n* = 1, HIV *n* = 2). The mean of the five measurements is shown for each in Table 1. Finally, the third quartile of all mean values of non-neoplastic CNS diseases measured in our study is 7.7% methylated CpG sites in the MGMT promoters. Mean methylation values above this threshold were defined as MGMT promoter hypermethylation. Individual samples among the groups of multiple sclerosis (*n* = 6/28), PML (*n* = 3/10), EPM/CPM (*n* = 6/8), and Wallerian degeneration (*n* = 2/3) showed significantly enhanced MGMT promoter methylation compared with controls. The median of each group is shown. The methylation status of *n* = 71 gliomas served as a comparison for the values measured here (see Appendix A).

**Figure 2 ijms-22-03845-f002:**
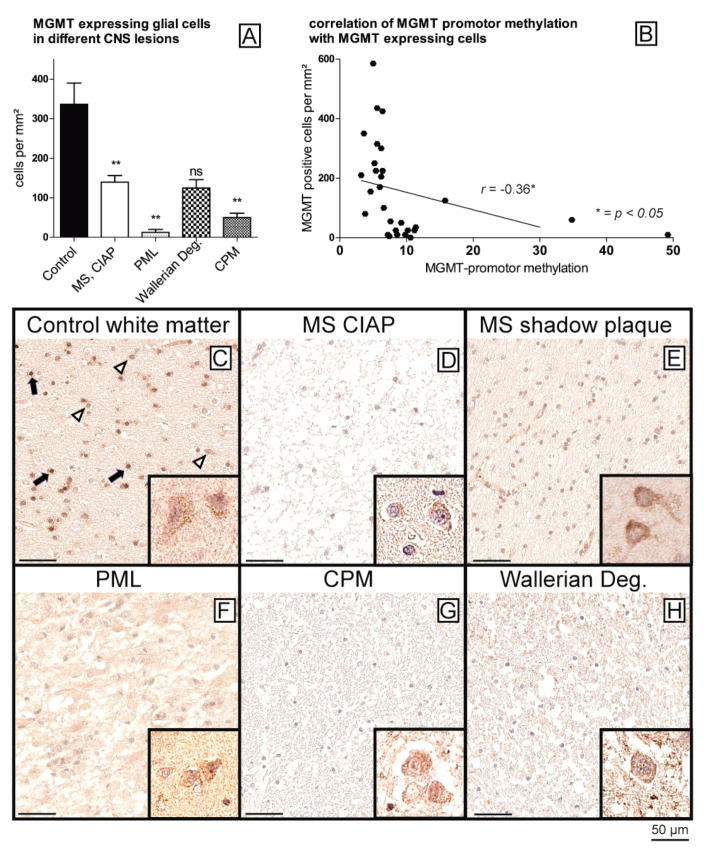
Reduction in MGMT protein expression in samples with enhanced MGMT promoter methylation. Compared to controls, all samples with enhanced MGMT promoter methylation show a reduction in MGMT protein expression in glial cells (** *p* < 0.01) (**A**). MGMT promoter methylation is negatively correlated with protein expression (**B**), Pearson, r = −0.36, *p* < 0.05). Immunohistochemistry of MGMT in white matter of controls (**C**) shows preserved MGMT expression in glial cells (astrocytes (arrowheads) and oligodendrocytes (arrows)), with MGMT staining appearing slightly stronger in oligodendrocytes (morphologically the slightly smaller cells with roundish nuclei). In all other conditions with increased MGMT promoter methylation, such as MS (**D**,**E**); PML (**F**); CPM (**G**), and Wallerian degeneration (**H**), reduced MGMT staining was detected in all white matter glial cells. The degree of methylation of the lesions did not play a role in the MS samples (**D**), chronic inactive plaque; (**E**), remyelinated plaque-shadow plaque). In all samples, the neurons in the same sections with a similar positive staining of MGMT served as an internal positive control to exclude staining artifacts (inset in (**C**–**H**) and Appendix A, edge length 15 µm each). Scale bar = 50 µm.

**Figure 3 ijms-22-03845-f003:**
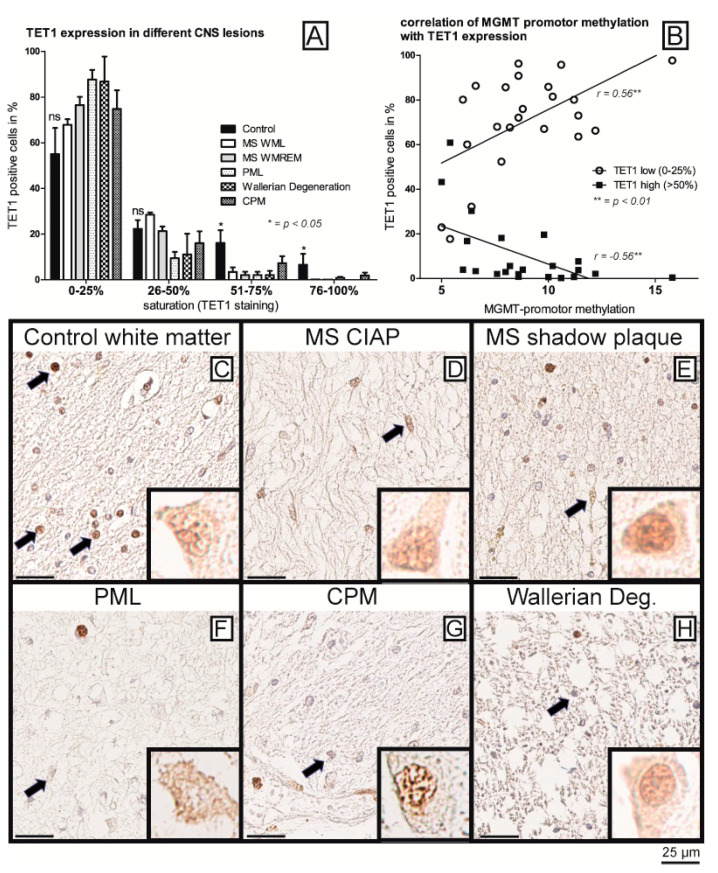
Reduction in TET1 staining in samples with enhanced MGMT promoter methylation. In control samples (**A**,**C**) a broad spectrum of TET1 expression was detectable with numerous strongly TET1-stained glial cells (arrows). In samples with enhanced MGMT methylation (**D**–**H**), glial cells with little or no TET1 expression were mainly detected, whereas strongly TET1-positive glial cells, i.e., glial cells with high TET1 expression were hardly detectable compared with controls (**D**). MS chronic inactive plaque, (**E**). MS shadow plaque; (**F**). PML, (**G**). CPM; (**H**). Wallerian degeneration). In all samples, the neurons in the same sections with a similar positive staining of MGMT served as an internal positive control to exclude staining artifacts (inset in (**C**–**H**), edge length 15 µm each). (**B**): The subset of glial cells with strong TET1 expression (lower panel) shows a significant negative correlation (Pearson r= −0.56, *p* < 0.01) with the level of MGMT promoter methylation, while vice versa the subset of glial cells with weak TET1 expression (upper panel) shows a significant positive correlation (Pearson r = 0.56, *p* < 0.01) with the level of MGMT promoter methylation. Scale bar = 25 µm.

**Figure 4 ijms-22-03845-f004:**
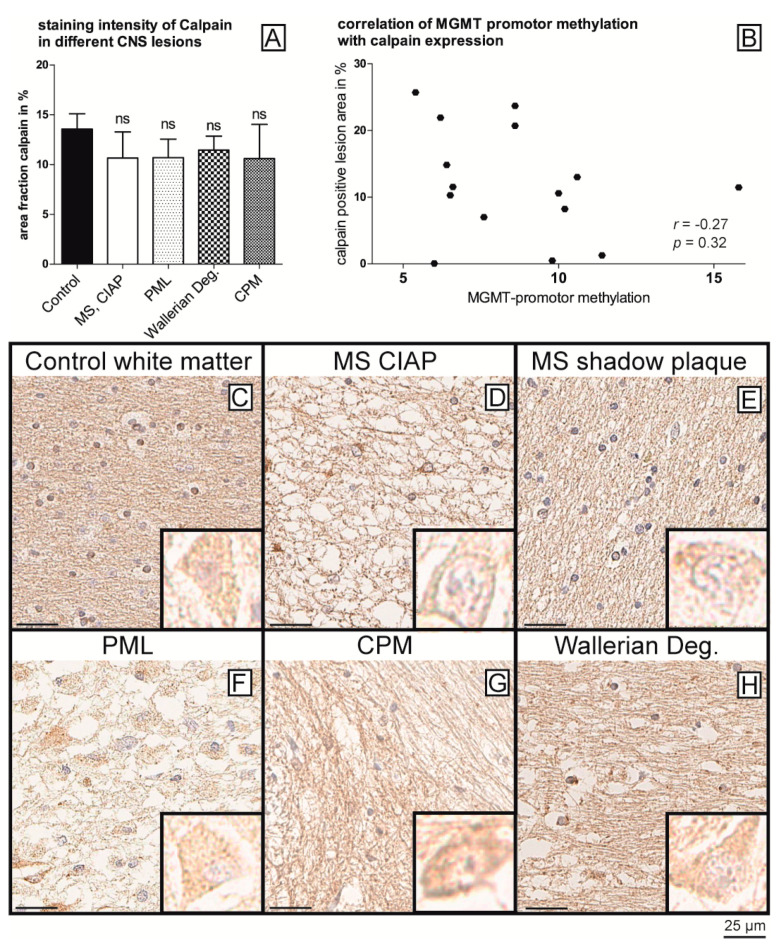
Calpain-1 staining of controls and samples with enhanced MGMT promoter methylation. Quantification (**A**) of Calpain-1 expression in glial cells reveals no significant differences in staining intensity between controls (**C**) and hypermethylated cases (MS—chronic inactive plaque (**D**); MS—shadow plaque (**E**); PML (**F**); CPM (**G**), or Wallerian degeneration (**H**)), neither in glial cells nor in neurons (inset in (**C**–**H**), edge length 15µm each). Consequently, no correlations were found between Calpain-1 expression and MGMT methylation (**B**). Scale bar = 25 µm.

**Table 1 ijms-22-03845-t001:** O^6^-methylguanine-DNA methyltransferase (MGMT) promoter methylation of the five measured CpG sites in the MGMT promoter.

Case						MGMT Promoter Methylation (%) at Different Positions					
	Age	Sex	Cause of Death	Disease Duration	Biopsy/Autopsy	Pos. 1	Pos. 2	Pos. 3	Pos. 4	Pos. 5	Mean (1–5)
Control											
control 1	75	f	multiorgan failure	na	autopsy	2	6	5	7	5	5
control 2	77	f	sepsis	na	autopsy	2	5	8	6	10	6.2
control 3	61	m	heart failure	na	autopsy	4	5	6	7	10	6.4
control 4	54	f	respiratory failure	na	autopsy	2	4	8	6	7	5.4
control 5	57	f	heart failure	na	autopsy	3	4	9	7	10	6.6
control 6	56	m	multiorgan failure	na	autopsy	2	5	7	4	12	6
control 7	58	m	heart failure	na	autopsy	3	4	5	4	4	4
control 8	78	m	heart failure	na	autopsy	4	5	6	5	5	5
Multiple Sclerosis											
MS autopsy 1	75	f	pneumonia		autopsy	5	5	9	7	16	8.4
MS autopsy 2	49	m	pneumonia		autopsy	2	4	4	2	4	3.2
MS autopsy 3	57	f	respiratory failure		autopsy	2	6	11	6	18	8.6
MS autopsy 4	52	m	multiorgan failure		autopsy	2	56	100	44	44	49.2
MS autopsy 5	68	f	pneumonia		autopsy	3	6	9	3	8	5.8
MS autopsy 6	62	f	cachexia and pulmonary insufficiency		autopsy	1	6	9	5	7	5.6
MS autopsy 7	44	m	multiorgan failure		autopsy	1	6	4	4	3	3.6
MS autopsy 8	57	f	respiratory failure		autopsy	9	15	10	8	14	11.2
MS autopsy 9	78	f	stroke		autopsy	9	10	12	12	14	11.4
MS autopsy 10	55	m	respiratory insufficiency complicating pneumonia and urosepsis		autopsy	4	6	7	5	10	6.4
MS autopsy 11	56	f	respiratory insufficiency in pneumonia		autopsy	3	6	4	6	7	5.2
MS autopsy 12	44	m	aspiration pneumonia		autopsy	2	4	9	7	9	6.2
MS autopsy 13	63	m	pneumonia		autopsy	6	9	5	6	10	7.2
MS autopsy 14	53	m	assisted suicide		autopsy	4	3	4	3	5	3.8
MS autopsy 15	54	f	heart failure		autopsy	3	4	5	2	7	4.2
MS autopsy 16	48	f	respiratory failure		autopsy	4	6	9	8	11	7.6
MS autopsy 17	58	m	terminal renal failure		autopsy	4	7	3	3	11	5.6
MS autopsy 18	66	f	cancer metastases in the liver resulting in severe failure of the liver functions		autopsy	3	4	5	3	8	4.6
MS autopsy 19	56	f	respiratory insufficiency in pneumonia		autopsy	6	7	9	7	8	7.4
MS autopsy 20	26	f	multiorgan failure		autopsy	6	8	9	8	15	9.2
MS biopsy 1	43	m	na	na	biopsy	3	4	5	3	6	4.2
MS biopsy 2	35	f	na	1 month	biopsy	3	5	4	2	5	3.8
MS biopsy 3	9	m	na	<1 month	biopsy	4	5	4	4	8	5
MS biopsy 4	25	f	na	na	biopsy	3	4	6	4	7	4.8
MS biopsy 5	31	m	na	na	biopsy	2	4	6	4	8	4.8
MS biopsy 6	46	f	na	<1 month	biopsy	2	3	3	3	3	2.8
MS biopsy 7	61	m	na	na	biopsy	4	5	5	4	4	4.4
MS biopsy 8	35	f	na	15 years	biopsy	3	3	4	3	3	3.2
PML											
PML 1	34	m	na	na	biopsy	4	6	8	4	6	5.6
PML 2	31	m	na	na	biopsy	6	6	7	6	8	6.6
PML 3	41	m	renal failure	na	autopsy	6	8	11	8	7	8
PML 4	51	m	na	na	autopsy	8	13	7	12	13	10.6
PML 5	65	f	na	na	autopsy	7	8	15	10	11	10.2
PML 6	58	f	na	na	autopsy	3	6	6	3	7	5
PML 7	77	f	na	na	biopsy	4	6	7	5	7	5.8
PML 8	66	f	na	na	biopsy	3	5	7	5	9	5.8
PML 9	41	m	renal failure	na	autopsy	4	6	8	6	12	7.2
PML 10	59	f	na	na	biopsy	3	4	5	4	4	4
CPM/EPM											
CPM 1	54	m	sepsis	na	autopsy	6	6	10	6	15	8.6
CPM 2	52	f	na	na	autopsy	5	8	6	7	17	8.6
CPM 3	54	m	sepsis	na	autopsy	5	7	10	7	12	8.2
CPM 4	52	f	na	na	autopsy	23	35	30	44	42	34.8
CPM 5	53	m	CPM, cerebral hemorrhage	na	autopsy	3	5	3	4	4	3.8
CPM 6	86	m	CPM	na	autopsy	7	6	7	5	3	5.6
CPM 7	41	m	CPM, cerebral hemorrhage	na	autopsy	3	7	10	9	10	7.8
CPM 8	55	m	stroke	na	autopsy	8	10	11	9	11	9.8
Wallerian degeneration											
WAL 1	59	m	multiorgan failure	na	autopsy	12	19	13	14	21	15.8
WAL 2	50	m	central regulatory failure	na	autopsy	5	7	12	7	13	8.8
WAL 3	65	m	stroke	na	autopsy	2	5	5	2	7	4.2
brain abscess											
ABS 1	42	m	na	na	biopsy	2	4	4	2	4	3.2
ABS 2	45	m	na	na	biopsy	3	4	5	4	5	4.2
ABS 3	3	f	na	na	biopsy	3	5	5	4	6	4.6
mycosis											
Myc 1	55	m	na	na	autopsy	2	4	5	4	5	4
Myc 2	48	f	na	na	biopsy	3	5	5	4	6	4.6
Myc 3	56	f	stroke, sepsis	na	autopsy	4	4	7	4	7	5.2
Myc 4	84	f	na	na	biopsy	4	5	5	4	6	4.8
toxoplasmosis						6	6	9	7	9	7.4
Toxo 1	59	f	na	na	biopsy	3	3	6	4	5	4.2
Toxo 2	47	m	na	na	biopsy	6	5	8	7	6	6.4
Toxo 3	64	m	na	na	biopsy	3	3	5	4	6	4.2
Toxo 4	29	f	na	na	biopsy	3	4	4	4	5	4
Toxo 5	40	m	na	na	biopsy	5	6	8	3	7	5.8
Toxo 6	22	m	na	na	biopsy	6	6	9	7	9	7.4
CMV											
CMV	38	m	na		autopsy	6	5	12	3	8	6.8
HSV											
HSV	37	m	HSV-encephalitis	4 weeks	autopsy	1	6	4	6	5	4.4
HIV											
HIV 1	44	m	respiratory failure	na	autopsy	4	7	9	5	10	7
HIV 2	51	m	multiorgan failure	na	autopsy	5	8	8	7	11	7.8

Abbreviations: m, male; f, female; na, not applicable; PML: progressive multifocal leucencephalopathy; CPM: central pontine myelinolysis; EPM: extrapontine myelinolysis.

**Table 2 ijms-22-03845-t002:** Antibodies and staining procedures.

Antigen	Company	Pre-Treatment	Dilution
MGMT, ab39253, mouse monoclonal	Abcam	EDTA	1:50
TET1, HPA019032, rabbit polyclonal	Sigma	citrate	1:200
TET2, ab94580, rabbit polyclonal	Abcam	citrate	1:100
Calpain1, ALS16293, goat polyclonal	BioMol	citrate	1:50
caspase3, 9661, rabbit polyclonal	Cell Signaling	citrate	1:200
SMI31, 801601, mouse monoclonal	Biolegend	EDTA	1:1000

## Data Availability

The data that support the findings of this study are available from the corresponding author upon reasonable request.

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
