# Peer review of "MGMT-Methylation in Non-Neoplastic Diseases of the Central Nervous System"

_ijms, 2021, doi:10.3390/ijms22083845_

Round 1
Reviewer 1 Report
The promoter methylation of 06-methylguanine-DNA methyltransferase (MGMT) has been shown to correlate with several diseases. However, in non-neoplastic brain diseases, promoter methylation of MGMT is not explored. In this manuscript, the authors aim to discover the correlation between promoter methylation of MGMT and non-neoplastic brain diseases. From their data, although an increasing trend is observed in MS, PML, EPM/ZPM, and Wallerian degeneration, the promoter methylation of MGMT shows no significant difference between healthy controls and disease patients. Next, they found the reduced MGMT protein is negatively correlated with enhanced MGMT promoter methylation. Finally, the authors claim that reduction of TET1 could be a factor in enhancing methylation of MGMT promoter in the disease patients. Overall, although the findings show some degree of interest and novelty, the data are preliminary, and no cause and effect experiment is carried out. Several points may help the authors to improve and strengthen their manuscript.
Major points:
- Overall, as the authors state, more samples need to be collected and analyzed to reach statistical significance.
- Authors only detect five positions of methylation in their manuscript. It is suggested to increase the examined position.
- Authors imply that the methylation of the MGMT promoter is associated with long chronic MS disease. To support their observation (or could be a hypothesis), examining methylation of MGMT promoter after treating glial cells with chronic factors may strengthen their finding (or hypothesis).
- Since the methylation of the promoter is the balance between DNA methyltransferases and demethylases, the validation of other potential factors should be included (for example, the expression of DNMTs).
- From Figure 2-4, the expression of MGMT, TET1, and Calpain-1 needs to be quantified in both glial cells and neurons. Please use arrows or arrowheads to point out representative cells in the images.
Minor points:
- Line 70, please replaces SEM to SD since SD is used throughout the manuscript.
- Although it is intuitive to interpret f, m, and na as female, male, and not applicable, respectively, it will be helpful if authors could write them in the table legend (lines 75-83).
- It will be good to see the sample size in the figure legend of Figure 1.
- For the data in lines 102-105, lines 136-143, lines 203-204, the authors could show them in the supplementary data.
- For the image labeling of figures 2-4, the letter c of control should be capital.
- In Figure 3A, what kind of cell type do the authors indicate?
Reviewer 2 Report
The authors describe their work on MGMT-methylation in non-neoplastic diseases of the CNS. It was found that slightly enhanced MGMT promoter methylation in individual cases of inflammatory demyelinating CNS diseases. It was concluded that a reduction in the expression of the demethylase TET1 may be a possible cause for enhanced MGMT promoter methylation. This is an interesting study. Appropriate methodology has been employed and the conclusions appear to be justified based on the data at hand. I have a few recommendations for consideration.
- Abstract. In the abstract the authors need to end this paragraph with a definitive statement on what clinical relevance and significance their findings have.
- Introduction. Please provide a clear hypothesis to be tested in the study.
- Results. For the IHC images, please indicate areas of interest with arrows.
- Results/Discussion. Is there any influence of sex and age as well as the cause of death on the extent of methylation?
- The authors need to emphasize and elaborate on the novelty aspect of their work as well as the clinical meaning and applicability of their findings.
i the
Round 2
Reviewer 1 Report
The authors have made a significant improvement in their manuscript. I appreciate all the great job they did to strengthen the manuscript. Although some of the wet lab experiments are not performed, I am also fine that they use literature reviewing instead and looking forward to seeing it to be published.
Before publishing, one more minor point (point 5) that the authors did not correct in the previous version: For the image labeling of figures 2-4, the letter c of control should be capital. Please correct them.
Author Response
We thank the reviewer again for the helpful criticism and useful comments in the first round of review.
Of course, we correct the last remaining minor point 5:
The c of controls has now been capitalized in figures 2-4.